

# Examining the assumptions of heterogeneity-based management for promoting plant diversity in a disturbance-prone ecosystem

Daniel J. McGlinn[1] and Michael W. Palmer[2]

[1] Department of Biology, College of Charleston, Charleston, SC, USA
[2] Department of Plant Biology, Ecology, and Evolution, Oklahoma State University, Stillwater, OK, USA

## ABSTRACT

**Background:** Patch-burn management approaches attempt to increase overall landscape biodiversity by creating a mosaic of habitats using a patchy application of fire and grazing. We tested two assumptions of the patch-burn approach, namely that: (1) fire and grazing drive spatial patch differentiation in community structure and (2) species composition of patches change through time in response to disturbance.
**Methods:** We analyzed species cover data on 100 m$^2$ square quadrats from 128 sites located on a 1 × 1 km UTM grid in the grassland habitats of the Tallgrass Prairie Preserve. A total of 20 of these sites were annually sampled for 12 years. We examined how strongly changes in species richness and species composition correlated with changes in management variables relative to independent spatial and temporal drivers using multiple regression and direct ordination, respectively.
**Results:** Site effects, probably due to edaphic differences, explained the majority of variation in richness and composition. Interannual variation in fire and grazing management was relatively unimportant relative to inherent site and year drivers with respect to both richness and composition; however, the effects of fire and grazing variables were statistically significant and interpretable, and bison management was positively correlated with plant richness.
**Conclusions:** There was some support for the two assumptions of patch-burn management we examined; however, in situ spatial and temporal environmental heterogeneity played a much larger role than management in shaping both plant richness and composition. Our results suggest that fine-tuning the application of fire and grazing may not be critical for maintaining landscape scale plant diversity in disturbance-prone ecosystems.

Corresponding author
Daniel J. McGlinn,
mcglinndj@cofc.edu

# INTRODUCTION

Natural variability concepts of land management are increasingly used in restoration ecology (*Palmer, Ambrose & Poff, 1997*; *Fuhlendorf et al., 2006*, *2009*; *Winter et al., 2012*).

Underlying these concepts are two premises: historical conditions and processes can provide guidance for management, and spatial and temporal variability generated by disturbance are vital components of many ecosystems (*Landres, Morgan & Swanson, 1999*). Proponents of natural variability concepts believe that managing for historical conditions will benefit species that have evolved in that system and will therefore minimize local extinctions of native taxa (*Swanson et al., 1994*; *Cissel, Swanson & Weisberg, 1999*). Additionally, spatial and temporal environmental heterogeneity is thought to maintain biological diversity (*MacArthur, 1965*; *Petraitis, Latham & Niesenbaum, 1989*).

In fire prone ecosystems it has been suggested that increasing the variability in the application of prescribed fire will increase habitat diversity and thus biological diversity (*Martin & Sapsis, 1992*; *Brockett, Biggs & van Wilgen, 2001*; *Fuhlendorf & Engle, 2001*, *2004*). Specifically, *Fuhlendorf & Engle (2001*, *2004)* suggested the fire-grazing feedback or pyric-herbivory, in which recently burned patches are preferentially grazed, could be used to create a *shifting mosaic* of habitat types when fire is patchily distributed on the landscape in contrast to the traditional homogeneous application of fire. They suggested that fire results in higher grazing intensity which promotes short-term dominance of forbs (high local species richness). As time since fire increases, grazing intensity decreases which allows grasses to become dominant and to competitively exclude forbs (low local species richness). They argue that a mosaic of burned and unburned patches more closely approximates the historical variability that would have existed in rangeland ecosystems and will result in higher landscape scale biodiversity than traditional homogeneous management practices. Their argument is essentially that "pyrodiversity begets biological diversity."

Three important assumptions underlie the application of patch-burn management to promote plant diversity:

I)  Fire and grazing drive spatial patch differentiation in species composition (high forb dominance to high grass dominance) and corresponding changes in richness,

II)  Species composition of local patches changes through time in response to frequency and time since disturbance, and

III)  High spatiotemporal variation in fire and grazing causes high compositional variation and thus high landscape-scale diversity

Tests of the patch-burn approach have thus far focused primarily on testing assumption III by comparing the compositional variability and diversity of management units where fire is either heterogeneously or homogeneously applied (*McGranahan et al., 2012*; *Kelly et al., 2012*; *Taylor et al., 2012*; *Farnsworth et al., 2014*). The large amount of literature on the effects of fire and grazing suggests that assumptions I and II should generally be true in plant communities. However, given that vegetation responds to many drivers (e.g., topo-edaphic heterogeneity, climate) simultaneously, it is unclear how important I and II are to explaining overall variability in grassland community structure and thus how effective the patch-burn approach is likely to be for the objective of promoting plant diversity.

Given that considerable resources are expended applying fire and grazing in rangeland systems to achieve management goals it is critical that we consider the importance of disturbances relative to in situ landscape heterogeneity in shaping community structure. Therefore, we used a long-term observational study at a preserve being managed with a patchy application of fire and free roaming cattle and bison grazers to test assumptions I and II of the patch-burn management approach in a tallgrass prairie plant community. We discuss the relevance of our results to the understanding and management of this disturbance prone ecosystem.

## MATERIALS AND METHODS

### Study site

We conducted our study at the Tallgrass Prairie Preserve (TGPP) which is a 15,700 ha nature preserve that is managed using a patch-burn approach (*Hamilton, 1996*, *2007*; *Allen et al., 2009*). The TGPP is located between 36.73° and 36.90° N latitude, and 96.32° and 96.49° W longitude, in Osage County, Oklahoma and owned by The Nature Conservancy (TNC). Over the course of the 12 year study period (1998–2009), total annual rainfall varied from 494 to 1,252 mm. The preserve is situated at the southern extent of the Flint Hills region. The elevation of the preserve ranges from 253 to 366 m, and the underlying bedrock of the region is characterized by soils derived from Permian limestone, sandstone, and shale (*Oviatt, 1998*). Due to the proximity of bedrock to the surface and the relatively steep terrain, the Flint Hills region has experienced long-term erosion leaving surface layers of soil that are thin and young. Because of this rockiness the Flint Hills region, including the TGPP, has remained unplowed and has been instead used primarily as rangeland for cattle. Prior to the acquisition of the preserve by TNC in 1989, the majority of the site was managed for cow-calf and yearling cattle production with a 4- to 5-year rotation of prescribed burning and patchy aerial application of broadleaf herbicides (1950–1989) (*Hamilton, 2007*).

Approximately 90% of the TGPP consists of grasslands. The majority of the grasslands are composed of tallgrass prairie habitats dominated by *Andropogon gerardii*, *Sorghastrum nutans*, *Sporobolus compositus*, *Panicum virgatum*, and *Schizachyrium scoparium*. Shortgrass prairie habitat occurs to a lesser extent on more xeric sites and is dominated by *Bouteloua* spp. Despite the application of herbicide earlier in the 20th century, the flora of the preserve appears relatively intact with a total of 763 species of vascular plants (to date) of which 12.1% are exotic (*Palmer, 2007*).

### Disturbance regime

The management plan at the TGPP encompasses a wide range of spatial and temporal variation in the application of prescribed fire and cattle or bison grazing (*Hamilton, 1996*, *2007*). In 1993, 300 bison were introduced year-round onto a 1,960 ha portion of the preserve. As the bison herd increased in size, the area allotted to the herd was increased eight times to an area of 8,517 ha by 2007 (Fig. 1A, 54% of preserve area). Initial bison stocking rates were increased in 1999 to 2.1 animal-unit months ha$^{-1}$ (see *Hamilton, 2007* for additional details). Within the bison unit, animals were allowed to range freely and

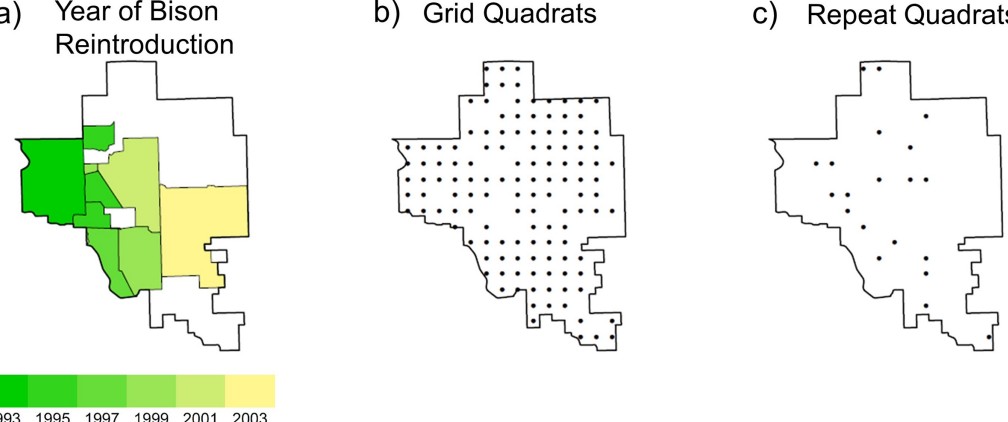

**Figure 1** **Maps of the study site the Tallgrass Prairie Preserve.** (A) The shaded area denotes the bison management area which increased in area during the study period (green region to yellow region), and the white area denotes areas grazed by cattle, (B) the 128 plots used for the grid analysis which were located on a 1 × 1 km UTM grid and were sampled primarily in 1997 and 1998, and (C) the locations of the 20 plots used for the repeat analysis which were sampled annually from 1998 to 2009.

their movement was not obstructed by internal fences. Watersheds within the bison unit were considered randomly for burning only if they met the minimum fuel criteria of 900 kg ha$^{-1}$ of fine fuels. Within a given year, the season of burn of the bison unit was split as follows: 40% dormant spring (March–April), 20% late growing season (August–September), and 40% dormant winter (October–December). The remainder of the preserve was seasonally grazed by cattle and typically burned more frequently in the dormant spring season, but some of the cattle pastures were utilized for smaller scale (2,350 ha) patch-burn experiments in which only one-third of a given management unit was burned annually (*Hamilton, 2007*). Stocking within the cattle pastures included both intensive-early stocking and season-long stocking, which contrasted with the year-round stocking in the bison unit.

## Data collection

During a 3-year period (1997–2000) we collected vegetation and environmental data at every intersection of a one km UTM grid for a total of 151 samples (Fig. 1B). For our analysis we only used 128 of the samples which had no standing water and less than 20% of cover by rocks or woody plants to ensure that they were representative of grasslands. We randomly selected 20 of these plots for annual resampling, which continued from 1998 to 2009 (Fig. 1C). All vegetation samples were collected in June (when we could readily identify both early and late-season plants). We used a plot design modified from *Peet, Wentworth & White (1998)*. At each site we recorded the presence/absence for plants identified to the species level that were rooted within each of the four corners of the 10 × 10 m quadrats at four nested spatial scales: 0.01, 0.1, 1, and 10 m$^2$ (see *McGlinn, Earls & Palmer, 2010* for a sampling diagram). At the 100 m$^2$ spatial scale we visually estimated species percent cover using cover classes (1: trace, 2: <1%, 3: 1–2%, 4: 2–5%, 5: 5–10%, 6: 10–25%, 7: 25–50%, 8: 50–75%, 9: 75–100%) rather than continuous

estimating cover to decrease the amount of error in our cover estimates (*Peet, Wentworth & White, 1998*). For the analysis, we used the mid-point of each cover class. For the purposes of this study we only report results at the 100 m² scale, but analyses at the finer spatial grains (using the corner subplots) yielded qualitatively similar results but higher levels of unexplained variance.

At every sampling event we combined four 15 cm soil cores collected at each corner of the plot. We sent the soil samples to Brookside Labs (New Knoxville, OH, USA) to be analyzed for soil cations: P, Ca, Mg, K, Na, B, Fe, Mn, Cu, Zn, and Al. The resampled vegetation and environmental data are in a public online archive (*McGlinn, Earls & Palmer, 2010*).

## Data analysis

Our analysis was composed of two sets of parallel analyses: those on species richness and those on species composition. Additionally, we carried out separate analyses on the 128 vegetation samples which were sampled once over a 3-year period and the 20 samples that were annually resampled over a 12-year period ($n = 20 \times 12 = 240$). We will refer to these separate analyses as the Grid analysis and the Repeat analysis, respectively. The Grid analysis was used to test assumption I and the Repeat analysis was used to test assumption II of the patch-burn approach.

Assumption I asserts that spatial variation in the community should reflect spatial variation in management. To carry out a strong test of this hypothesis we compared the effect sizes and variance explained by management variables relative to inherent environmental heterogeneity captured by soil cations on plant community richness and composition. We quantified variation in soil using the first three axes of a principal components analysis on the 11 soil cations (see Table S1 for cation PC loadings). We used three axes because these explained the majority of the variation in the dataset (68%) and this provided a balanced comparison of three soil variables against three management variables. We chose to examine soil variables because previous research at the site indicated that they are good proxies for spatially relevant environmental variation (*Palmer et al., 2003*).

Assumption II asserts that local patches change through time in response to disturbance. To test assumption II we examined the degree to which spatio-temporal changes in richness and composition correlated with changes in management after controlling for inherent site and year specific differences. While the Grid analysis examined the effects of key soil variables on species composition, it is possible that unmeasured variables that distinguish one site from another are responsible for intersite differences. Similarly, interannual variation may be inadequately summarized by readily obtainable climatic variables. Thus, the simplest way to completely account for interplot and interyear differences, independent of management, is to use site and year nomial variables, respectively.

In all the analyses, we quantified management using three variables: years of bison management, years since burn, and number of burns in the past 5 years. Year of bison management is meant to capture variation attributed to the differences in bison and cattle

management at the preserve. Note that sites not grazed by bison were grazed by cattle and there were no ungrazed sites. We chose not to include season of burn as an explanatory variable because 83% (67 out of 80) of the prescribed fire events recorded on our study sites took place during the dormant season.

In both the Grid and Repeat analyses, we quantified the variance explained by the competing classes of variables using variation partitioning which estimates the unique and shared fractions of explained variance in species richness and composition (*Peres-Neto et al., 2006*; *Legendre & Legendre, 2012*). The univariate species richness analyses were carried out using ordinary least squares regression (OLS). We also carried all of our analyses out using generalized least squares (GLS) regression that accounted for spatial and temporal autocorrelation in the error of our models (*Pinheiro & Bates, 2000*), but we found that the GLS model had qualitatively similar results to the OLS model. Therefore, we only report results from the OLS analysis below.

We analyzed species composition using redundancy analysis (RDA) and canonical correspondence analysis (CCA) direct ordination approaches. We only report the results of the RDA analyses because they were qualitatively similar to the CCA results. For the compositional analyses, we down-weighted the effect of common species using a square root transformation of the cover estimates. Following the recommendations of *Peres-Neto et al. (2006)*, we report the adjusted coefficient of determination $R^2_{adj}$ for each fraction of variance using Ezekiel's adjustment *Ezekiel (1930)* for the OLS and RDA analyses. We tested if the individual fractions of variation were significantly different than zero using permutation tests with 999 permutations of the rows of the response variable of interest.

All analyses were conducted in R (*R Development Core Team, 2018*) and the multivariate analyses were carried out using the package vegan (*Oksanen et al., 2018*). The code and data are publically available at an online repository (https://zenodo.org/record/2641111#.XLUvR-hKhnI). The data shared at this repository includes the data shared by *McGlinn, Earls & Palmer (2010)*, two-additional years of data, and the vegetation, management, and environmental data for the remainder of the plots on the UTM grid.

## RESULTS

### Grid analysis—test of spatial management effects

The OLS model explained 17% of the total variance in richness ($F_{6,121} = 4.13$, $p < 0.001$), and the RDA model explained 16% of the total variance in species composition ($F_{6,121} = 3.76$, $p < 0.001$). Spatial variation due to differences in soil properties explained the largest proportion of variance in species richness and composition (Fig. 2, $R^2_{OLSadj} = 0.15$ and Fig. 3, $R^2_{RDAadj} = 0.08$, respectively). The first soil PC axis reflected a gradient from high iron sites to high calcium sites (Table S1), and it was the most strongly correlated variable with richness (standardized partial regression coefficient, $\beta = -0.38$, $p < 0.001$, Table S2) and composition (Table S3). The independent effect of management was negligible on species richness (Fig. 2, $R^2_{OLSadj} \cong 0$, $p = 0.59$) but not on species composition (Fig. 3, $R^2_{RDAadj} = 0.02$, $p = 0.001$).

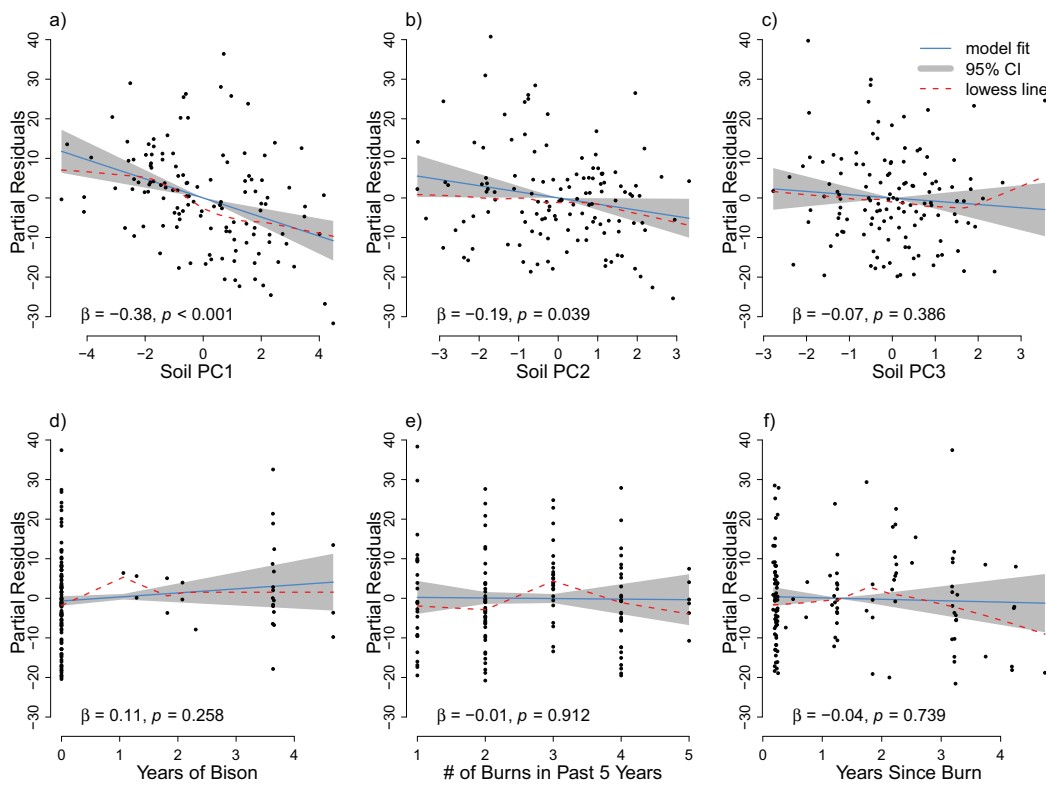

**Figure 2 The partial residuals of species richness relative to each soil PC axis (A–C) and each management variable (D–F) after controlling for the other explanatory variables considered in the grid analysis.** The OLS model is represented by the solid blue line with the gray 95% confidence interval. A lowess smoother is provided as a dashed red line. The standardized partial regression coefficient (β) and the associated $p$-value are reported for each explanatory variable.

## Repeat analysis—test of space-time management effects

For the repeat analysis, the OLS model explained 76% of variance in richness ($F_{33,206} = 19.23$, $p < 0.001$), and the RDA model explained 61% of the total variance in species composition ($F_{33,206} = 9.78$, $p < 0.001$). We observed statistically significant management effects on both richness (Fig. 4) and composition (Fig. 5); however, relative to site and year effects management explained little of the variance ($R^2_{\mathrm{OLSadj}} = 0.04$ and $R^2_{\mathrm{RDAadj}} = 0.01$ for richness and composition, respectively, Fig. 6). The majority of variance in richness and composition was attributed to site specific effects ($R^2_{\mathrm{OLSadj}} = 0.49$ and $R^2_{\mathrm{RDAadj}} = 0.45$, respectively, Fig. 6). Year effects were intermediate to those of site and management effects. The majority of the variance attributed to management was shared with site effects (Fig. 6). Years of bison management was the most important management variable (Figs. 4 and 5; Tables S4 and S5) which was positively correlated with richness (β = 0.46, $p < 0.001$, Fig. 4). Years of bison management increased cover of many weedy annual plants such as *Amphiachyris dracunculoides* (amphidrac), *Kummerowia stipulacea* (kummstip), *Chamaecrista fasciculata* (chamfasc), *Medicago lupulina* (medilupu), and *Melilotus officinalis* (melioffi). Bison management was correlated with decreased cover of several long-lived perennials species such as *Helianthus mollis*

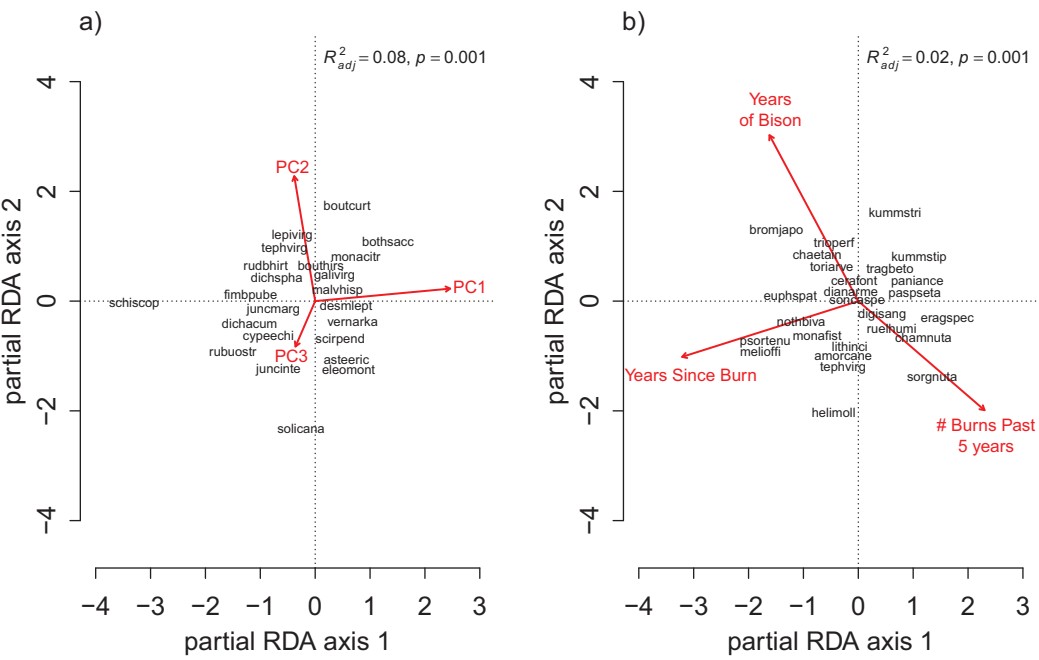

**Figure 3 The partial RDA biplots for species composition in the grid quadrats as explained by (A) the soil PC variables after controlling for the management variables, and (B) the management variables after controlling for the soil PC variables.** The explanatory variables are red arrows and the species codes designate the species scores on the first two ordination axes. Species codes are only provided when they are legible prioritized by their degree of fit to the ordination axes. The species codes are defined in the online data repository: https://github.com/mcglinnlab/tgp_management/blob/master/data/species_traits.csv.                

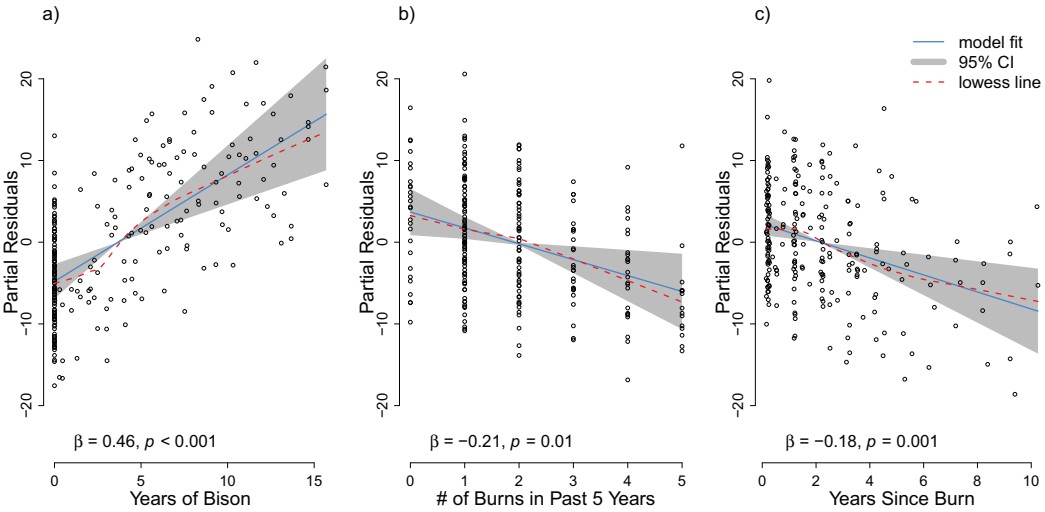

**Figure 4 Partial residuals of richness for the management variables (A) years of bison management, (B) the number of burns in the past five years, and (C) years since last burn after controlling for site and year effects in the repeat analysis.** The OLS model is represented by the solid blue line with the gray 95% confidence interval. A lowess smoother is provided as a dashed red line. The standardized partial regression coefficient (β) and the associated *p*-value are reported for each explanatory variable.                

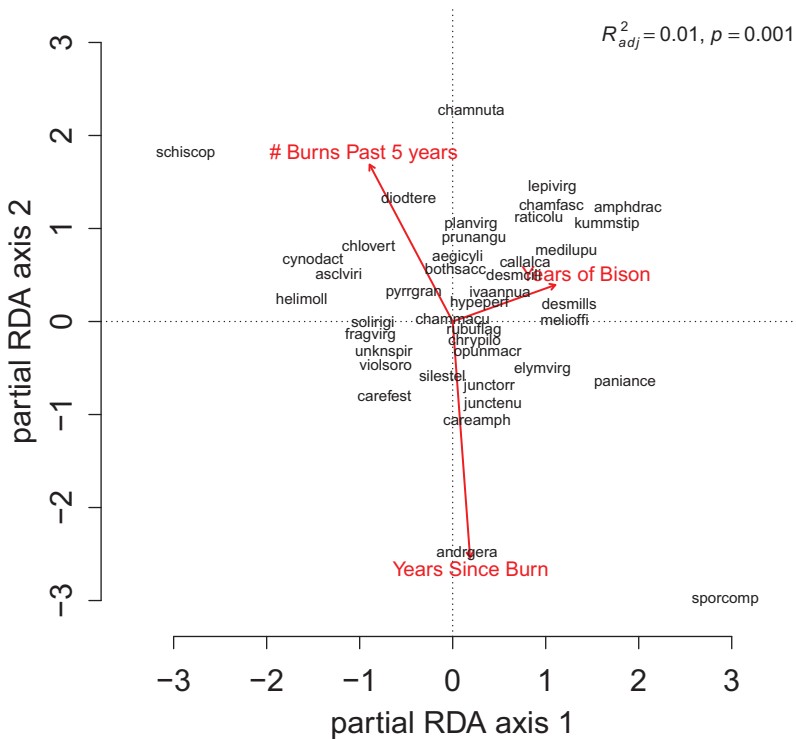

**Figure 5 The partial RDA biplot for species composition in the repeat quadrats as explained by management variables after controlling for site and year effects.** Symbols and species codes as defined in Fig. 3.

(helimoll), *Asclepias viridis* (asclviri), *Spiranthes* spp. (unkspir), and *Viola sororia* (violsoro) (Fig. 5). The number of burns in the past 5 years and time since the last burn were both significantly negatively correlated with species richness (Fig. 4). Frequent burning increased *Schizachyrium scoparium* (schizscop) but decreased *Andropogon gerardii* (andrgera) and *Sporobolus compositus* (sporcomp) (Fig. 5).

## DISCUSSION

We found that species richness and composition of a tallgrass prairie plant community were relatively insensitive to changes in the disturbance regime related to fire frequency and grazer management. Inherent sources of heterogeneity swamped out the influence of management related variables. The Grid analysis demonstrated that management was correlated with spatial variation in species composition (not richness), but that soil heterogeneity due to a gradient (soil PC1) from sandstone derived soils (high iron sites) to limestone derived soils (high calcium sites) explained a much larger portion of overall variance in both richness and composition (Fig. 6). The Repeat analysis demonstrated that management influenced the spatio-temporal variation in vegetation as posited by the patch-burn paradigm; however, this was a rather trivial portion of total explained variance in both richness and composition with respect to proportion of inherent spatial and temporal variation (i.e., independent site and year effects, Fig. 6). Our results suggest that variation in the application of fire and grazer management should only be expected to

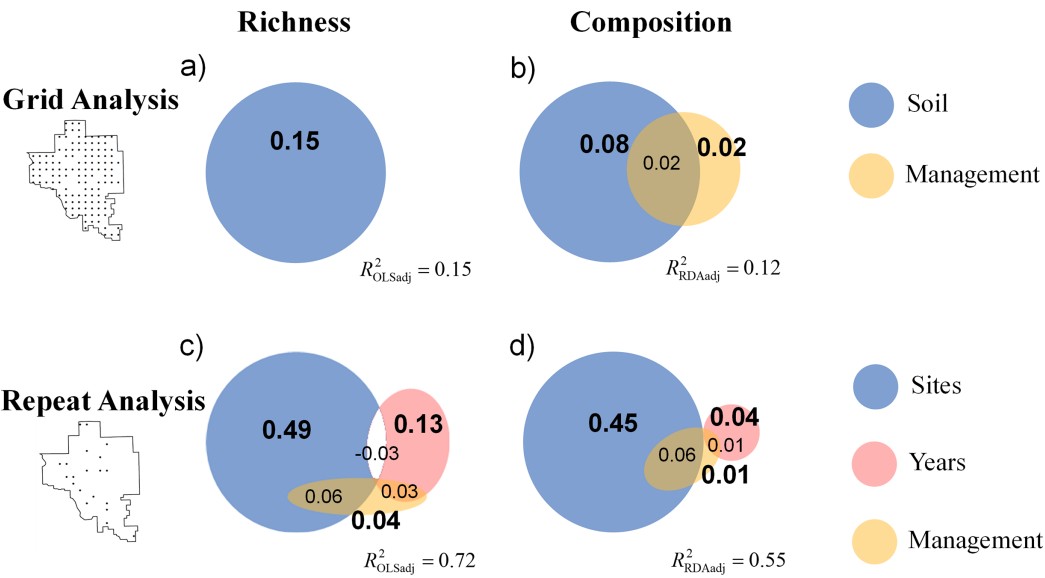

**Figure 6 Venn diagrams display the results of variation partitioning for (A) the grid analysis on richness, (B) the grid analysis on composition, (C) the repeat analysis on richness, and (D) the repeat analysis on composition.** The grid analysis examines if management drives spatial variation in community structure, and the repeat analysis examines if management drives a spatio-temporal changes in vegetation. The relative area of the circles and reported statistics represent the adjusted explained variance for richness (from OLS) and for composition (from RDA) attributed to either the soil or management variables. The percentages in bold are statistically significant ($p = 0.001$). A negative value of shared explained variance is presented as an unshaded region. The overall adjusted explained variance and is reported for each analysis. Note that management explained approximately zero variance in richness in (A).

have modest effects on plant biodiversity relative to inherent sources of landscape heterogeneity. We did not directly test the effectiveness of patch-burning to increase plant diversity, but our findings suggest that patch burning should not be expected to have large effects on plant richness and composition due to the overwhelming importance of soil and climate drivers.

The spatial variation in vegetation was best explained by a soil gradient from low nutrient, sandstone derived soils to higher nutrient, limestone derived soils. Several other studies have emphasized the importance of soil properties on tallgrass prairie vegetation and in particular the role of soil nutrients (*Turner et al., 1997*; *Burke et al. 1998*; *Baer et al., 2003*, *2005*). Although we did not directly measure soil nutrients in this study, *Brokaw (2004)* demonstrated at the Repeat quadrats that soil Ca was positively correlated with total N and residual P. Higher levels of nutrients in the more calcareous sites could have resulted in stronger competitive exclusion and may provide an explanation for why these sites had lower richness (*Grime, 1973*; *Tilman, 1982*). Alternatively, *Pärtel's (2002)* species pool hypothesis postulates that negative correlations between richness and pH (or calcium given their tight correlation) may be explained by the fact that calcareous sites were less frequent when the flora was evolving. *Palmer et al. (2003)* found that the data from the TGPP (including this study's sites as well as many others) appeared to support *Pärtel's (2002)* hypothesis in the grasslands but not in the

woodlands. Two of the dominant grasses differed strongly in their abundance along this soil gradient: *Schizachyrium scoparium* (little bluestem) was more abundant on the acidic sites, and *Sporobolus compositus* (composite dropseed) was more abundant on the calcareous sites which is consistent with the basic natural history of these species.

Although management effects were trivial, they were interpretable. Bison management was linked to increases in richness and increased cover of weedy annuals. This may be due to the wallowing behavior of bison (*Knapp et al., 1999*). Richness decreased with more frequent fire suggesting that too much fire disturbance can result in homogenization of the community (*Fuhlendorf & Engle, 2001*, *2004*). Additionally, we found that too long of a time since the last burn decreased plant richness and increased cover of two dominant warm-season grasses *Andropogon gerardii* and *Sporobolus compositus* which is also consistent with *Fuhlendorf & Engle's (2001*, *2004)* hypothesis that pyric-herbivory increases forb diversity by decreasing competitive exclusion by dominant grasses.

Our results regarding the relatively minor role of management-based disturbances are consistent with a variety of studies that have found mixed support for the premise that "pyrodiversity begets biological diversity" in tallgrass prairie biomes. A regional scale study on three different groups of insects (ants, butterflies, and leaf beetles) in tallgrass prairies found no differences in diversity due to patchy verses homogenous management (*Debinski et al., 2011*). *McGranahan et al. (2012)* found that management for heterogeneity increased variance in grassland vegetation structure and life-form (e.g., forb, grass, woody) composition at the scale of a burn patch in only three out of five sites along a precipitation gradient. Additional analyses of the same five sites uncovered that various habitat variables responded inconsistently in response to heterogeneity based management (*McGranahan et al., 2013*). We note that their analysis included our study site, but they did not examine the response of species richness or composition. Also at our same study site, two studies on avian communities found that some metrics of landscape scale bird diversity were slightly but significantly higher in pach-burn treatments while other metrics showed no change (*Coppedge et al., 2008*; *Hovick et al., 2015*). *McGranahan et al. (2018)* examined a range of different patch-burn management designs at our study site and found that spatial and temporal heterogeneity in plant functional groups and vegetation structure was highest in pastures with intermediate numbers of patches. *Gibson & Hulbert (1987)* also examined the relative importance of fire, topography, and climate on tallgrass prairie vegetation but in contrast to our results they found time since fire to be the most important variable followed by topography. This difference may be because our study site is 4.5 times larger than their study area, and thus encompassed more spatial environmental heterogeneity. It seems likely that the role of disturbance is more detectable when the environment is more homogenous.

At a global extent, there has also been mixed evidence that pyrodiversity increases biodiversity (*Bowman et al., 2016*; *Kelly & Brotons, 2017*). Although a full review of all the pyrodiversity literature is beyond the scope of the current work, here we note a few important findings. *Tingley et al. (2016)* found that diversity was strongly increased by pyrodiversity in montane bird communities, and *Ponisio et al. (2016)* demonstrated

plant and pollinator diversity increased with increased pyrodiversity. However, studies in the Mallee region of south-eastern Australia found no support that management using greater pyrodiversity resulted in higher diversity of small mammals (*Kelly et al., 2012*), birds (*Taylor et al., 2012*), or lizards (*Farnsworth et al., 2014*). *Kelly et al. (2015)* synthesized the results of the Mallee fire studies and argued that some mosaic of patch ages is required for optimal conservation. Ant communities in Kruger National Park and in an Australian tropical savanna were also largely insensitive to the fire regime (*Parr & Andersen, 2006*; *Andersen et al., 2014*). *Beale et al. (2018)* carried out the first continental-scale analysis of pyrodiversity effects, and they found that across Africa pyrodiversity in wet but not dry savannas was positively correlated with mammal and bird species richness. They suggested that the importance of managing to increase pyrodiversity may depend on the climate regime of the system.

The results of our study suggest that fine-tuning the application of disturbance may be of less importance given the apparent stability of plant community structure to disturbance and the potential importance of other sources of inherent landscape heterogeneity on plant species composition and richness (*Parr & Andersen, 2006*; *Andersen et al., 2014*). This may be welcome news for land managers because it suggests that disturbance-prone communities may be relatively resilient to uncertainty in the prescription of disturbance such as fire frequency and grazer choice. Our results also suggest that management plans attempting to harness pyric-herbivory in the context of a patch-burn management scheme to increase community heterogeneity may take time to show a strong effect and that the underlying environmental template should be considered when developing these plans. We agree with *McGranahan et al. (2013)* that given the high degree of uncertainty in the response of communities to attempts to increased habitat heterogeneity, management decisions should focus on specific conservation priorities rather than simply the increase of habitat heterogeneity.

We observed that the influence of spatiotemporal variation in fire and grazing on plant richness and composition was relatively minor relative to inherent variation between sites in a relatively intact tallgrass prairie. Nevertheless, variation in fire and grazing had significant effects consistent with two basic assumptions of the patch-burning management approach. Given the overriding influence of inherent landscape heterogeneity on the plant community as well as results from other taxonomic groups and systems suggests that fine-tuning the application of disturbances may not be crucial for maintaining plant biodiversity.

## ACKNOWLEDGEMENTS
A. Ellison, D. Bowman, S. Winter, B. Allred, D. Schoolmaster, K. Riemer, P. Adler, E. White, Z. Roehrs, M. Allen, S. Fuhlendorf, and several anonymous reviewers provided comments that improved the quality of this manuscript. The U.S. EPA has not officially endorsed this publication, and the views expressed herein may not reflect the views of the Agency. We thank Bob Hamilton, members of Laboratory for Innovative Biodiversity and Analysis, and the Osage Nation and numerous other researchers for assisting us at various stages in the field.

### Funding

Daniel J. McGlinn received funding from the U.S. Environmental Protection Agency (EPA) under the Greater Research Opportunities Graduate Program. Michael W. Palmer received support from NSF Grant Number EPS-0447262 and EPS-0919466, The Oklahoma State University College of Arts and Science, The Oklahoma Nature Conservancy, The Spatial and Environmental Information Clearinghouse, The Philecology Trust, The Oklahoma Water Resources Research Institute. The funders had no role in study design, data collection and analysis, decision to publish, or preparation of the manuscript.

### Grant Disclosures

The following grant information was disclosed by the authors:
U.S. Environmental Protection Agency (EPA) under the Greater Research Opportunities Graduate Program.
NSF: EPS-0447262 and EPS-0919466.
The Oklahoma State University College of Arts and Science, The Oklahoma Nature Conservancy, The Spatial and Environmental Information Clearinghouse, The Philecology Trust, The Oklahoma Water Resources Research Institute.

### Competing Interests

The authors declare that they have no competing interests.

### Author Contributions

- Daniel J. McGlinn conceived and designed the experiments, performed the experiments, analyzed the data, contributed reagents/materials/analysis tools, prepared figures and/or tables, authored or reviewed drafts of the paper, approved the final draft.
- Michael W. Palmer conceived and designed the experiments, performed the experiments, contributed reagents/materials/analysis tools, authored or reviewed drafts of the paper, approved the final draft.

### Data Availability

Data is available at GitHub: https://github.com/mcglinnlab/tgp_management.

### Supplemental Information

Supplemental information for this article can be found online at http://dx.doi.org/10.7717/peerj.6738#supplemental-information.

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
