# Peer review of "Examining the assumptions of heterogeneity-based management for promoting plant diversity in a disturbance-prone ecosystem"

_PeerJ, doi:10.7717/peerj.6738_

## Round 0.1 · original submission · Major Revisions

Both reviewers were supportive of publishing this paper, and I concur.

Key items to address in your revision:

1. Per reviewer 1. You are testing assumptions, but the title ("foundations") and discussion are somewhat more wide-ranging. You should be a bit more modest in what you accomplish here.

2. Per reviewer 2. I understand the confusion this reviewer expresses in the comments about assumption 2. One part of confusion is that in the abstract, you state that assumption is that "species composition of patches diverge through time in response to disturbance". (note: should be "diverges"). But in the text, you restate the assumption as: "Species composition of local patches changes through time in response to frequency and time since disturbance". The latter, which is essentially an assumption of successional processes, seems straightforward to address in the way you have done. But the former (in the abstract) is very different. Patches changing through time is very different from patches diverging through time. As I'm not 100% familiar with the pyrodiversity and management literature, I don't know which is correct. But you are testing the latter phrasing of the assumption, not the former. Either way, you need to ensure that the assumptions in litt., your text, and the analyses are congruent.

3. Figures 2 and 3 are interesting illustrations of variance partitioning, but I would also like to see plots of the actual data. Please include those in the revision.

4. Please carefully proofread the revision.

·

Basic reporting

The paper is based on a thorough analysis of a data set, that is available for re-examination, using state-of-the-art statistical techniques. The English expression is of a high quality and the citations are appropriate.

Experimental design

The design of the experiment is appropriate to address the very narrow question of the relative importance of site and disturbance (grazing and fire) ion plant species composition and richness.

Validity of the findings

My concern with this paper is that the study does not tackle the fundamental question in pyrodiversity and pyric herbivory – how spatial scale of burning affects grazing, and if there is a self-reinforcing feedback with post fire grazing through changed fuel arrays. To do this would require information of the effect of fire patch size on grazing and fire hazard (flammability) of vegetation.

So this paper makes a useful point about the importance of underlying physical gradients in shaping plant species distribution and plant community composition but I do not believe it directly tests the core ideas of pyrodiversity and pyric herbivory. The Discussion needs, therefore, to be more balanced in light of the very narrow focus of this study objectives and design.

Additional comments

I would scale back the claims of this paper as being a major test of pyric herbivory and pyrodiversity, rather it looks at some facets of these complicated ideas. The authors should reflect on the caveats with their study and thereby temper their recommendations for managers. More thought about the biological processes leading to the results would strengthen this paper (i.e. why do some species prosper under some disturbance and edaphic regimes while other species do not).

Reviewer 2 ·

Basic reporting

Overall, well written with scattered minor typos. Meets all the criteria listed by PeerJ.

Experimental design

Seems fine - nice use of long-term vegetation data

Michael Palmer is the go-to guy for multivariate analysis of vegetation data, in particular, and I suspect Daniel McGlinn knows his stuff, as well.

Validity of the findings

See general comments below.

Additional comments

This manuscript describes an analysis of the effectiveness of “pyroherbivory” in maintaining species diversity in grass-grazer ecosystems. The authors test two assumptions: 1) fire and grazing create spatial variability in community structure, and 2) species composition of patches diverge through time in response to these disturbances. Using long-term data from permanent vegetation sampling plots, they find that underlying heterogeneity in soil chemistry, as a proxy for abiotic variability, had a much larger impact on vegetation structure and variability than did patch grazing, as did interannual variation in precipitation. However, patch grazing did have some slight positive effect on species richness. They conclude that “fine tuning” a patch grazing management scheme may provide limited benefits to maintain grassland plant species richness.

This is a very nicely written and easy to follow manuscript that addresses some interesting aspects of the patch-grazing model for rangeland management. This pyroherbivory approach was actually developed and promoted by colleagues of the authors at Oklahoma State, and it has been widely promoted as a way to enhance species diversity without negatively affecting cattle production. The effectiveness of this approach certainly deserves analysis. The patch-grazing model does have strong underlying logic. Traditional grazing approaches promote uniform grazing across a pasture, which logically should result in lower spatial heterogeneity and lower species diversity at the pasture scale. Patch grazing, on the other hand, driven by a rotational burning of subsets of a pasture each year to create different aged patches that are grazed to different degrees will increase heterogeneity and thus pasture scale species diversity. The approach has been tested in many systems on various taxa, and as the authors point out evidence for the success of this approach is mixed.

In the case of the current manuscript the authors find that underlying physical environment has a larger effect than the management scheme, although patch grazing did have a statistically small positive effect. As far as I can tell from the literature and that reviewed in this manuscript, the pyroherbivory approach results in either neutral or positive impacts to some degree, but it never has a negative effect, so in that regard it seems reasonable to suggest the approach has some benefits when implemented, but do the costs of that management scheme justify the benefits? Is the focus on the statistical contribution in your model a valid assessment of whether or not this approach is useful in practice?

Where I am a bit confused by this manuscript is assumption (2) that says patches should diverge in composition over time. Either I don’t understand the pyroherbivory scheme or that assumption is not correct. The authors need to better explain how that assumption works since it is one of the 2 main questions in this manuscript. Here’s my confusion. Again, if a manager burns the entire pasture every year and spreads water and salt licks throughout, then uniform grazing will occur and heterogeneity will be low. Patch burning, for example, would divide the pasture into six units, two of which would be burned each year essentially creating 6 patches that are all burned every three years. Within any given year the treatments would create spatial heterogeneity and fire refugia because 1/3 of the area would be burned that year, 1/3 would be 1 year post burn and 1/3 would be 2 years post burn. Over time all sections would be subjected to the same 3-year fire rotation and grazing pressure. So why should they diverge? Within any given year the three patch types should diverge in community structure to some degree.

The manuscript has scattered typos and a more careful proof reading is needed next time. For example, under methods in the abstract it says “in the grasslands of the preserve” which up to that point neither grassland nor preserve have been mentioned….

---

## Round 0.2 · accepted · Accept

Thank you for attending to the revisions. You've done a great job, and it will be good to see this paper published.

#